# Accelerating Diffusion LLMs via Adaptive Parallel Decoding

**Daniel Israel**
Department of Computer Science
University of California, Los Angeles
disrael@cs.ucla.edu

**Guy Van den Broeck**
Department of Computer Science
University of California, Los Angeles
guyvdb@cs.ucla.edu

**Aditya Grover**
Department of Computer Science
University of California, Los Angeles
adityag@cs.ucla.edu

## Abstract

The generation speed of current LLMs is bottlenecked by autoregressive decoding, where tokens are predicted sequentially one by one. Alternatively, diffusion large language models (dLLMs) theoretically allow for parallel token generation, but in practice struggle to achieve the speed of autoregressive models without significantly sacrificing quality. We therefore introduce adaptive parallel decoding (APD), a novel method that dynamically adjusts the number of tokens sampled in parallel. We achieve this by defining a multiplicative mixture between the dLLM marginal probabilities and the joint probability of sequences under a small auxiliary autoregressive model. This inverts the standard setup of speculative decoding, where the goal is to sample from a large autoregressive verifier by drafting from a smaller model. We further optimize APD by enabling KV caching and limiting the size of the masked input. Altogether, our method puts forward three tunable parameters to flexibly tradeoff throughput and quality. We show that APD provides markedly higher throughput with minimal quality degradations on downstream benchmarks.

## 1 Introduction

Large language models (LLMs) have remarkable text generation capabilities and have attracted evergrowing interest and widespread adoption. However, a significant impediment to their deployment lies in the speed of text generation [43]. The dominant paradigm, autoregressive models [32], generates tokens one by one in a sequential manner. While this approach has yielded state-of-the-art results in terms of quality, the inherent sequentiality of generation limits throughput and hinders real-time applications, especially as models continue to scale in size [19]. In addition, the recent phenomenon of test-time scaling [35] and reasoning models [13] suggest that generation speed will be an important bottleneck for increasing LLM capabilities. This challenge has spurred research into alternative approaches beyond autoregressive sequential sampling.

One promising alternative to purely sequential generation is offered by diffusion large language models (dLLMs) [42]. These models, inspired by successes in image generation [36], theoretically permit the parallel generation of multiple tokens simultaneously, offering a path towards significantly faster inference. In this work, we challenge the assumption that dLLMs, exemplified by the open-source models Dream [41] and Llada [29], can be practically used for parallel generation without additional modifications. We find that the quality of these models is best achieved by generating tokens one at a time (one timestep per token), and attempts to exploit the parallelizability of diffusion

**Prompt**

Toulouse has twice as many sheep as Charleston. Charleston has 4 times as many sheep as Seattle. How many sheep do Toulouse, Charleston, and Seattle have together if Seattle has 20 sheep?

**Autoregressive Decoding**

Let's start by defining the number of sheep each city has in terms of variables and then use the given information to find the exact numbers.
1. Let $S$ represent the number of sheep in Seattle.
2. According to the problem, Seattle has 20 sheep, so $S = 20$.
3. Charleston has 4 times as many sheep as Seattle. So, let $C$ represent the number of sheep in Charleston:
$$C = 4S = 4 \times 20 = 80$$
4. Toulouse has twice as many sheep as Charleston. Let $T$ represent the number of sheep in Toulouse:
$$T = 2C = 2 \times 80 = 160$$
5. To find the total number of sheep in Toulouse, Charleston, and Seattle together, we add the number of sheep in each city:
$$T + C + S = 160 + 80 + 20 = 260$$
Therefore, the total number of sheep in Toulouse, Charleston, and Seattle together is 260.

Latency: 7.08 seconds        Throughput: 37 tokens per second

**Adaptive Parallel Decoding**

Let's break down the information:
* Seattle has 20 sheep.
* Charleston has 4 times as many sheep as Seattle, so Charleston has 4 * 20 = 80 sheep.
* Toulouse has twice as many sheep as Charleston, so Toulouse has 2 * 80 = 160 sheep.
Now, let's add up the number of sheep each city has:
Seattle: 20 sheep
Charleston: 80 sheep
Toulouse: 160 sheep
Total number of sheep: 20 + 80 + 160 = 100 + 160 = 260
So, Toulouse, Charleston, and Seattle have a total of \boxed{260} sheep together

Latency: 2.75 seconds        Throughput: 59 tokens per second

Figure 1: **Autoregressive vs Adaptive Parallel Decoding (APD).** We visualize a sample from Qwen2.5 7B decoded autoregressively (baseline) and Dream 7B with APD (our method). Contiguous text of the same color indicates tokens sampled in parallel. Here, we show that APD generates the solution much faster than autoregressive sampling, which is sequential and slow.

models suffer a reduction in quality. Thus, current state-of-the-art dLLMs, namely Dream and Llada as currently conceived, fail to match the speed and quality of autoregressive LLMs.

This gap between theoretical and practical performance creates an opportunity for novel decoding mechanisms that can effectively harness the parallel generation capabilities of dLLMs while maintaining high fidelity to the target text distribution. The core challenge is that when sampling multiple tokens in parallel, one only has access to the marginal distribution of each token, which ignores inter-token dependencies [23]. A decoding algorithm that increases parallelism when sampling from dLLM and maintains generation quality must consider the joint distribution that captures these dependencies.

To address this challenge, we introduce Adaptive Parallel Decoding (APD), a novel decoding algorithm designed to dynamically modulate the number of tokens sampled in parallel during the generation process with dLLMs. In APD, we first fix the generation order of the dLLM to be left to right. This unexpected change makes the dLLM autoregressive, but we empirically show that it maintains generation quality and in some cases can improve it. By using the dLLM autoregressively, we can then also use a smaller autoregressive model to determine which subset of tokens sampled in parallel adequately captures joint dependence. Our criterion depends on a multiplicative mixture between the diffusion model and autoregressive model. Note that while speculative decoding uses a large model to verify samples from a small draft model, our task is to use a smaller verification model to check the quality of a larger model. This dynamic leads to a fundamentally new problem statement.

Our contributions extend beyond the conceptual framework of APD. We further optimize the decoding process by incorporating practical improvements to diffusion model sampling such as Key-Value (KV) caching, a technique traditionally associated with autoregressive models [31]. We also gain significant speed by limiting the size of the masked input to the model. These simple changes substantially increase the computational efficiency of diffusion model inference.

Collectively, we introduces three distinct tunable parameters. These parameters provide practitioners with the flexibility to navigate the inherent trade-off between generation throughput and output quality, allowing for tailored configurations that can meet diverse application-specific requirements. This paper will demonstrate that Adaptive Parallel Decoding offers a significant step forward in accelerating dLLM inference (and LLM inference in general). We present empirical evidence showing that APD achieves substantially higher throughput compared to existing LLM decoding strategies, all while incurring only minimal degradations in quality across a range of downstream benchmark tasks. The subsequent sections will discuss the technical details of APD, present a comprehensive set of experiments validating our claims, and discuss the broader implications of our findings for the future of efficient LLM generation.

## 2    Background

In the following sections, we will define notation to be used throughout the paper. We consider a data point as a sequences of $n$ tokens $x = (x_1, ..., x_n)$. For sets of indices $\mathcal{Q}, \mathcal{O} \subseteq \{1, ..., n\}$ for which $\mathcal{Q} \cap \mathcal{O} = \emptyset$, a masked language model $p_{\mathrm{D}}(\cdot \mid \cdot; \theta)$ computes the marginal probabilities of tokens with query indices $\mathcal{Q}$ conditioned on tokens with observed indices $\mathcal{O}$.

$$p_{\mathrm{D}}(x_{\mathcal{Q}} \mid x_{\mathcal{O}};\ \theta) = \prod_{i \in \mathcal{Q}} p_\theta(x_i \mid x_{\mathcal{O}}) \tag{1}$$

where $p_\theta$ is a learned conditional distribution parameterized by $\theta$. In this work, a diffusion language model is a masked language model, and these terms can be used interchangeably.

### 2.1    Discrete Diffusion Language Models

Discrete diffusion models [26] have recently emerged as a promising alternative to traditional autoregressive approaches for language generation, offering benefits such as non-autoregressive generation capabilities and inherent iterative refinement [39]. Discrete diffusion models are masked language models [34] trained to reverse a data corruption process $q$ that stochastically converts a clean sequence of tokens $x^0$ to a noisy $x^t$, gradually converting clean tokens to [MASK] over time $t$

$$q_{t|0}(x_i^t \mid x_i^0) = \begin{cases} t, & \text{if } x_i^t = [\text{MASK}] \\ 1 - t, & \text{if } x_i^t = x_i^0 \\ 0 & \text{otherwise} \end{cases} \qquad q_{t|0}(x^t \mid x^0) = \prod_i q_{t|0}(x_i^t \mid x_i^0) \tag{2}$$

Given this noise process, dLLMs are trained to maximize a lower bound on the log-likelihood computed using conditional distributions of clean data [30].

$$\log p_\theta(x^0) \geq \mathbb{E}_{t \sim U(0,1), x^t \sim q(x^t \mid x^0)} \frac{1}{t} \left[ \log p_{\mathrm{D}}(x_{\mathbb{1}(x^t = [\text{MASK}])} \mid x_{\mathbb{1}(x^t \neq [\text{MASK}])}; \theta) \right] \tag{3}$$

While dLLMs are trained in theory to randomly unmask tokens at inference time, in practice decoding heuristics are used to determine the order of tokens to unmask [20]. For example, to achieve the best results, Dream unmasks tokens according to lowest entropy as assessed by the diffusion model at each timestep. Llada performs the best when unmasking according to the highest probability, which is referred to as "confidence" based decoding [29].

In light of these inference time modifications, diffusion language models are able to compete with autoregressive model in terms of generation quality. However, in realistic scenarios, the quality of a generation must be weighed against the speed. When we consider both of these factors, current open-source dLLMs are far from competitive with their open-source autoregressive counterparts. In Table 1, we find that dLLMs can perhaps achieve competitive quality, but their throughput is a fraction of Qwen2.5 7B [40]. Another observation is that in order to achieve competitive performance on GSM8K [6], dLLMs must generate 256 tokens in 256 timesteps, which is *actually sequential*. We see that decreasing the number of timesteps can increase throughput at the expense of quality. Nonetheless, the throughput will not approach autoregressive speed without drastic reduction in quality. Interestingly, we find the simplest decoding order, just left to right, yields good results in most scenarios, and in the case of GSM8K the best results.

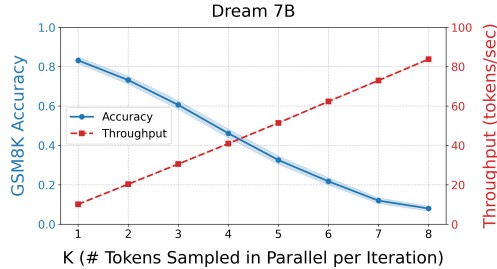 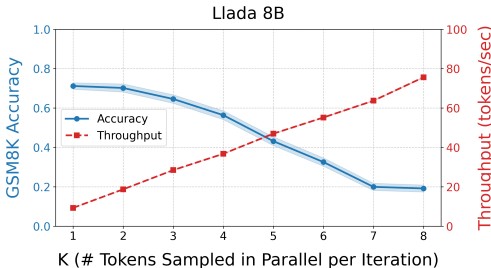

Figure 2: **Naive Parallel Generation.** Fixing the number of tokens $k$ to semi-autoregressively generate (left to right) in parallel per iteration, we observe a clear tradeoff between parallelization and generation quality. This tradeoff exists with open source diffusion models: Dream and Llada.

## 2.2 From Sequential to Parallel Sampling

In this work, we utilize diffusion models with a particular noise schedule: denoising left to right. This is equivalent to sampling autoregressively and can be used to compute an exact autoregressive likelihood.

$$p_{\text{AR}}(x;\theta) = \prod_{i=1}^{n} p_{\text{D}}(x_i \mid x_{<i};\theta) \tag{4}$$

To further parallelize this process, one can decode multiple tokens at a time semi-autoregressively [38] from groups of size $k$. Decoding in this manner samples from the following distribution

$$p_{\text{SAR}}(x;\theta,k) = \prod_{i=1}^{\lfloor (n-1)/k \rfloor + 1} p_{\text{D}}(x_{(ik-k+1:ik)} \mid x_{<(ik-k+1)};\theta) \tag{5}$$

where it follows that $p_{\text{AR}}(x;\theta) = p_{\text{SAR}}(x;\theta,1)$ for $k=1$. A diffusion model can parallelize the term inside the product by sampling from each marginal distribution independently. Although this will lead to faster generations, the intra-group independence assumption is poor and leads to lower quality generations. We observe this empirically in Figure 2: as we increase the number of tokens sampled at a time, throughput increases, but downstream accuracy decreases. Thus, there exists a tradeoff between parallelization and speed with sample quality. However, instead of fixing $k$, our core insight is that the number of tokens to generate in parallel can be chosen adaptively. In this work, we argue that doing so significantly reduces the sharp tradeoff between speed and quality.

## 3 Method

### 3.1 Problem Statement

Our goal is to sample from groups $\mathcal{G} = \{(s_1, e_1), (s_2, e_2), ..., (s_l, e_l)\}$ where $(s_i, e_i)$ is a tuple that denotes start and end indices inclusive and $e_i + 1 = s_{i+1}$ for all $i$. Note that Equation 5 is a special

Table 1: dLLM Quality and Throughput with Different Decoding Approaches

| Model | GMS8K Accuracy | Throughput (tokens/sec) |
|---|---|---|
| Dream 7B (Random, 256 Steps) | $0.404 \pm 0.021$ | $3.31 \pm 0.068$ |
| Dream 7B (Entropy, 128 Steps) | $0.708 \pm 0.020$ | $7.57 \pm 0.157$ |
| Dream 7B (Entropy, 256 Steps) | $0.804 \pm 0.017$ | $4.28 \pm 0.080$ |
| Dream 7B (Left to Right, 256 Steps) | $0.832 \pm 0.016$ | $10.1 \pm 0.015$ |
| Llada 8B (Random, 256 Steps) | $0.456 \pm 0.022$ | $5.07 \pm 0.168$ |
| Llada 8B (Confidence, 128 Steps) | $0.526 \pm 0.022$ | $13.6 \pm 0.284$ |
| Llada 8B (Confidence, 256 Steps) | $0.534 \pm 0.022$ | $6.63 \pm 0.143$ |
| Llada 8B (Left to Right, 256 Steps) | $0.712 \pm 0.020$ | $9.33 \pm 0.016$ |
| Qwen2.5 7B (Autoregressive) | $\mathbf{0.854 \pm 0.015}$ | $\mathbf{38.6 \pm 0.004}$ |

case where $\mathcal{G} = \{(1, k), (k + 1, 2k + 1), ..., (n - k, n)\}$ We shall sample from

$$p_{\text{APD}}(x; \theta, \mathcal{G}) = \prod_{(s,e) \in G} p_{\text{D}}(x_{s:e} \mid x_{<s}) \tag{6}$$

In choosing $\mathcal{G}$, we must balance the following two goals — (1) *Speed*: minimize $|\mathcal{G}|$ and (2) *Quality*: minimize the distance between $p_{\text{APD}}$ and $p_{\text{AR}}$.

Achieving only one out of two goals is trivial. For maximum speed, we can sample from all the marginals of the diffusion model in one shot such that $\mathcal{G} = \{(1, n)\}$ and $|\mathcal{G}| = 1$. This would result in a significant drop in quality. In fact, the drop in quality can be quantified by mutual information

$$I_{p_{\text{AR}}}(x; \theta) = \text{KL}(p_{\text{AR}}(x_{1:n}; \theta) \;||\; p_{\text{D}}(x_{1:n}; \theta)) \tag{7}$$

Alternatively, we can set $\mathcal{G} = \{(1, 1)...(n, n)\}$, in which case $|\mathcal{G}| = n$ and $p_{\text{APD}} = p_{\text{AR}}$. This is very slow because it maximizes the number of sequential iterations. For architectural reasons to be discussed in Section 3.3, it is also very slow to sample autoregressively from a diffusion model because it cannot perform KV caching. This architectural difference distinguishes our problem setting from speculative decoding [22], because computing an autoregressive likelihood in a diffusion model is slow and sequential, whereas in an autoregressive model it is fast and parallelizable. While speculative decoding does not map onto our problem, we shall argue that *adaptive parallel decoding* achieves both goals together.

## 3.2   Adaptive Parallel Decoding

In adaptive parallel decoding, we assume to have access to a small autoregressive model $\hat{p}_{\text{AR}}$ that can compute the likelihood of sequences in parallel. While $p_{\text{D}}$ only computes marginal probabilities over tokens, $\hat{p}_{\text{AR}}$ computes a joint probability, allowing it to model dependencies between tokens. For brevity, we shall omit model parameters $\theta$.

To select $\mathcal{G}$, we shall focus our analysis on a subproblem. Given samples $\hat{x}_t, ..., \hat{x}_n \sim p_{\text{D}}(\cdot \mid x_{<t})$, we must select $k$ such that $\hat{x}_{t:t+k}$ are close to $p_{\text{AR}}$. We do not have access to $p_{\text{AR}}$ because evaluating its likelihood for each token will give us the worst-case speed. Instead, we have access to a small, inaccurate model $\hat{p}_{\text{AR}}$ that can quickly evaluate likelihood in parallel. To approximate $p_{\text{AR}}$, we shall define a target distribution which we use to determine how many tokens $k$ are accepted. This target distribution $p_T$ should satisfy the following desiderata:

(1) if $p_{\text{D}}(X = c) = 1$ and $\hat{p}_{\text{AR}}(X = c) < 1$, then $p_T(X = c) = 1$
(2) if $p_{\text{D}}(X = c) < 1$ and $\hat{p}_{\text{AR}}(X = c) = 1$, then $p_T(X = c) = 1$

Property (1) follows from the fact that $p_{\text{AR}}(x_{1:k}) \geq \sum_{i=1}^{k} p_{\text{D}}(x_i) - k + 1$ by Bonferroni's inequality [10], which means if $p_{\text{D}}(x_i) = 1$ for all $i$, then $p_{\text{AR}}(x_{1:k}) = 1$. Put verbally, if the marginals given by the diffusion model are 1, its joint distribution or $p_{\text{AR}}$ will also have probability 1. Property (2) is difficult to justify formally because we do not have a relationship between $p_{\text{AR}}$ and $\hat{p}_{\text{AR}}$. Assuming $\hat{p}_{\text{AR}}$ reasonably approximates the joint distribution, property (2) is a desirable heuristic because a token with joint probability 1 should be accepted.

A multiplicative mixture of distributions, also known as a product of experts [16], fulfills the two requirements above. We define the multiplicative mixture of $p_{\text{D}}$ and $\hat{p}_{\text{AR}}$ as follows

$$p_T(x) = \frac{1}{Z} p_{\text{D}}(x)^R \hat{p}_{\text{AR}}(x)^{1-R} \tag{8}$$

where $Z$ is the normalizing constant. Our mixture is defined in terms of a hyperparameter $R \in [0, 1]$. When it is high, it gives the diffusion model more weight. We may now give an accept criteria based on target distribution $p_T$. We adopt an accept criteria similar to existing parallel sampling methods based on universal coupling [1]. Formally, a universal coupler is a function $g$ which for a distribution $p$ and source of randomness $r$ satisfies

$$\mathbb{P}_{r \sim U[0,1]}(g(p, r) = x) = p(x) \tag{9}$$

For categorical distributions, the Gumbel-Softmax Trick [18] is a universal coupler [1]. An important property of a universal coupler is that for two distributions $p, p'$, their samples from a universal

---
**Algorithm 1** Adaptive Parallel Decoding
---
1: **Input:** Diffusion model $p_D$, Autoregressive model $\hat{p}_{AR}$, Mixture Weight Parameter $R$, Maximum sequence length $n$
2: **Output:** Generated token sequence $x$
3: $x \leftarrow ()$                                                 ▷ Stores the accepted tokens
4: $t \leftarrow 1$                                                    ▷ Index of token to generate
5: **while** $t \leq n$ **do**
6:     $\texttt{marginal\_logits}_{t:n} \leftarrow p_D(x_{t:n} \mid x_{<t})$
7:     $r \leftarrow \text{Gumbel}(0,1)$
8:     $\hat{x}_{t:n} \leftarrow \texttt{sample\_gumbel}(\texttt{marginal\_logits}_{t:n}, r)$
9:     $\texttt{joint\_logits}_{t:n} \leftarrow \hat{p}_{AR}(\hat{x}_{t:n} \mid x_{<t})$
10:    $\texttt{product\_logits}_{t:n} \leftarrow \texttt{softmax}(R * \texttt{marginal\_logits}_{t:n} + (1-R) * \texttt{joint\_logits}_{t:n})$
11:    $\hat{y}_{t:n} \leftarrow \texttt{sample\_gumbel}(\texttt{product\_logits}_{t:n}, r)$
12:    $k \leftarrow \texttt{sum}(\texttt{cumprod}(\hat{x}_{t+1:n} = \hat{y}_{t+1:n})) + 1$
13:    $x \leftarrow \texttt{concat}(x, \hat{x}_{t:t+k-1})$                           ▷ Append accepted tokens
14:    $t \leftarrow t + k$
15: **end while**
16: **return** $x$
---

coupler with a shared source of randomness are likely to be the same given that the distributions $p$ and $p'$ are similar [21].

$$\mathbb{P}_r(g(p,r) \neq g(p',r)) \leq 2\,\text{TV}(p,p') \tag{10}$$

where TV is total variation distance. Thus, in our algorithm we sample from the diffusion model using the Gumbel-Softmax trick as a universal coupler $g$ and source of randomness $r$: $\hat{x}_t, ..., \hat{x}_n \sim g(p_D, r)$. We then sample from our target: $\hat{y}_t, ..., \hat{y}_n \sim g(p_T, r)$, and we accept all tokens that are the same between $\hat{y}_i$ and $\hat{x}_i$ until the first disagreement. Because the goal is to sample from $p_{AR}$, we always accept the first proposed token $\hat{x}_t$. The full algorithm is given in Algorithm 1. While the same algorithm is valid even when using different Gumbel randomness to sample from the target and proposal distributions, intuitively using the same randomness will maximize the number of accepted tokens.

Algorithm 1 provides a sampling procedure which takes as input a diffusion model $p_D$, a small autoregressive model $\hat{p}_{AR}$, and tunable parameter $R \in [0,1]$. When $R = 1$, the target distribution is $p_D$ and the algorithm will accept every token from the diffusion model in one shot. When $R = 0$, the algorithm does not "trust" the diffusion model and instead only accepts tokens that $\hat{p}_{AR}$ accepts.

Our sampling algorithm is agnostic to the implementation of $p_D$ and $\hat{p}_{AR}$, but in practice architectural optimizations are used for additional speed. For example, when computing the joint logits of the autoregressive model, we can use KV caching [31] to avoid redundant computation on $t$ previous tokens. The only requirement for $p_D$ is that it can draw samples from the marginal distributions of the suffix in parallel. However, if we consider that the implementation of $p_D$ is a Transformer-based masked language model, further speedups can be achieved.

### 3.3 Recompute KV Window

In our above method, we center sampling parallel as the primary way to improve decoding speed. However, overall we aim to maximize the throughput of diffusion models by all means necessary, so we also must consider architectural factors. We enable KV caching for tokens outside a sliding window of size $W$. For a detailed explanation of KV caching, see Appendix B. Although KV caching with a diffusion model trained with bidirectional masking can induce arbitrary out-of-distribution behavior, empirically we observe very little performance degradation (Appendix B). The intuitive reason is that tokens sufficiently far from the rightmost token will have small attention weight, so inaccuracies in their KV will not dramatically change the overall attention computation.

### 3.4 Maximum Masked Lookahead

Because we are sampling from a diffusion model left to right autoregressively, we can exploit the fact that the input will contain a large block of contiguous [MASK] tokens as the suffix. The simple fix is

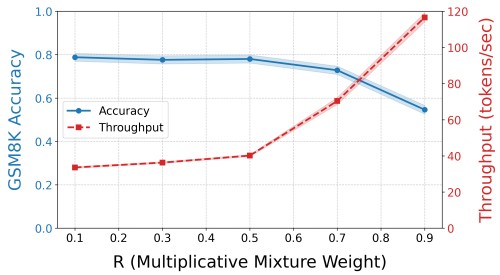 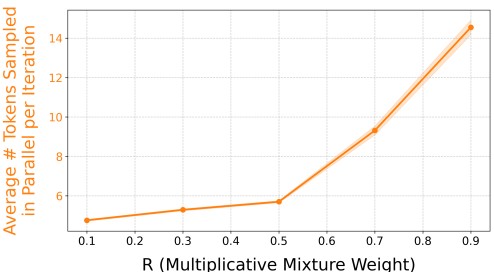

Figure 3: **Multiplicative Mixture Weight.** In Adaptive Parallel Decoding (APD), smaller $R$ values result in fewer parallel tokens per iteration but maintain high quality. In particular, we achieve a far better tradeoff compared to naively parallelizing semi-autoregressively. Notably, generating over 5 tokens per iteration on average is possible with APD while maintaining $\sim$80% accuracy on GSM8K. At the expense of some quality, even over 100 tokens per second is possible.

to set a maximum length $M$ for this suffix. Because the attention computation has $O(n^2)$ complexity, decreasing the size of the input can lead to significant speed improvements. Unfortunately, the size of the masked lookahead $M$ can change the output distribution by changing the probability of the end of sentence token [EOS]. Empirically, the maximum masked lookahead $M$ can be tuned to gain significant speedup with minimal quality loss.

## 4 Experiments

In our method, we define three tunable parameters, which we will briefly summarize:

1. Multiplicative Mixture Weight $R$: Higher results in higher throughput, lower quality
2. Recompute KV Window $W$: Lower results in higher throughput, lower quality
3. Maximum Masked Lookahead $M$: Lower results in higher throughput, lower quality

Our experimental goal is to empirically analyze the tradeoff in speed and generation quality as these parameters vary.

### 4.1 Implementation

In our experiments, we use Dream 7B Instruct [41] as the diffusion model $p_{\mathrm{D}}$ and Qwen2.5 0.5B [40] as the approximate autoregressive model $\hat{p}_{\mathrm{AR}}$. Both models have demonstrated impressive capabilities on math, science, and reasoning benchmarks. Conveniently, Dream 7B is a diffusion model that has been distilled from Qwen2.5 7B, so it is more likely to have distributional overlap with Qwen2.5 0.5B. These models satisfy another prerequisite; they share the same tokenizer. Although our method is theoretically applicable to Llada 7B, which exhibits strong left to right generation capabilities, Llada 7B is trained with a non-standard tokenizer that no existing autoregressive model utilizes.

### 4.2 Experimental Configuration

For the following experiments, we load the models in BF16 precision and run them on single NVIDIA 24GB A5000 GPU connected to a Colfax CX41060s-EK9 4U Rackmount Server with AMD EPYC (Genoa) 9124 processors. We operate using the LM Evaluation Harness [11] standard implementation of benchmarks with a few modifications and evaluate on GSM8K [6], GPQA [33], and MATH [15], and HumanEval [4]. See Appendix C for precise details.

### 4.3 Tradeoffs

For analyzing tradeoffs, we vary each parameter in isolation to measure the impact of each on performance. In later experiments, we plot configurations of the parameters jointly. Each plot shows the Grade School Math 8K (GSM8K) [6] accuracy with 500 samples, and the throughput measured by

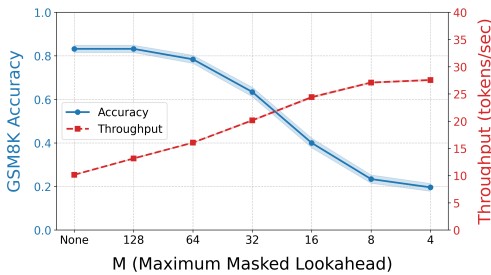 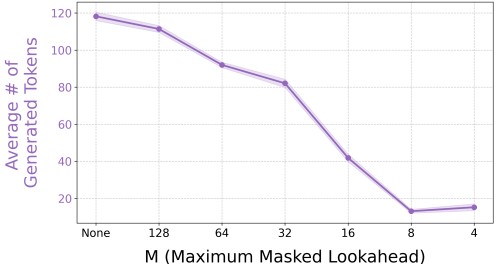

Figure 4: **Maximum Masked Lookahead.** We illustrate the trade-off between speed and quality when adjusting the maximum masked lookahead ($M$). Decreasing $M$ can significantly increase throughput (speed) but it will also reduce quality by shortening the generation length. Thus, this parameter should not be made excessively small.

number of tokens generated divided by generation time. We measure the standard error of throughput and accuracy with respect to the variation in samples, i.e. different math problem.

Figure 3 shows that as we vary the multiplicative mixture weight $R$, we achieve a range of speed, quality outcomes. We see for small $R$, we accept fewer tokens per iteration but maintain high quality. Remarkably, it is possible to generate over 5 tokens an iteration on average and achieve close to the same accuracy as generating 1 token per iteration. As we increase $R$, the accuracy drops but not drastically. Constrast this with Figure 2, where increasing the number of tokens sampled in parallel leads to a precipitous drop in generation quality. With our method, by dynamically choosing when to sample from the diffusion model in parallel, we are able to achieve much higher token acceptance rates without losing generation quality. In Table 2, we track statistics of APD and find that the parallel acceptance rate is high.

We also analyze the impact of changing the recompute KV window $W$ for left to right autoregressive generation from the diffusion model, sampling with with $p_{AR}$ (the same as $K = 1$). In Figure 6 (Appendix B), we observe the same relationship between accuracy and throughput, but the tradeoff is not as strong. As we decrease the window $W$, we can achieve a nontrivial speedup at almost no expense to quality.

Finally, we examine the speed-quality throughput exhibited by altering the maximum masked lookahead $M$. We again observe that there is no free lunch that can increase speed with zero degradations in quality. In this case, we find that decreasing $M$ increases throughput but can significantly alter quality by virtue of the fact that the generation length decreases. In general, works have shown a relationship between increased "thinking time" (the number of generated tokens) and reasoning strength [27]. Thus, changing the output distribution of $p_D$ in this way can damage generation quality, especially for complex reasoning tasks.

All together, the parameters in question each exhibit a unique tradeoff between speed and quality. We believe giving the user or practitioner more flexibility to balance these factors at inference time is an undoubted strength of our approach.

### 4.4 Pareto Frontier

The Pareto frontier characterizes the set of optimal configurations when faced with the inherent trade-off between two or more conflicting objectives, which in the case of LLM inference is speed and quality. A specific LLM configuration is considered Pareto optimal [25] if it is impossible to improve its performance in one objective without simultaneously incurring a detrimental effect on

Table 2: **APD Generation Statistics.** Mean number of tokens sampled in parallel.

| Model | GMS8K | GPQA | MATH |
|---|---|---|---|
| Dream 7B ($R = 0.7, W = 16, M = 100$) | $7.62 \pm 0.14$ | $4.35 \pm 0.03$ | $6.45 \pm 0.04$ |
| Dream 7B ($R = 0.6, W = 32, M = 200$) | $5.75 \pm 0.05$ | $4.31 \pm 0.02$ | $6.33 \pm 0.03$ |

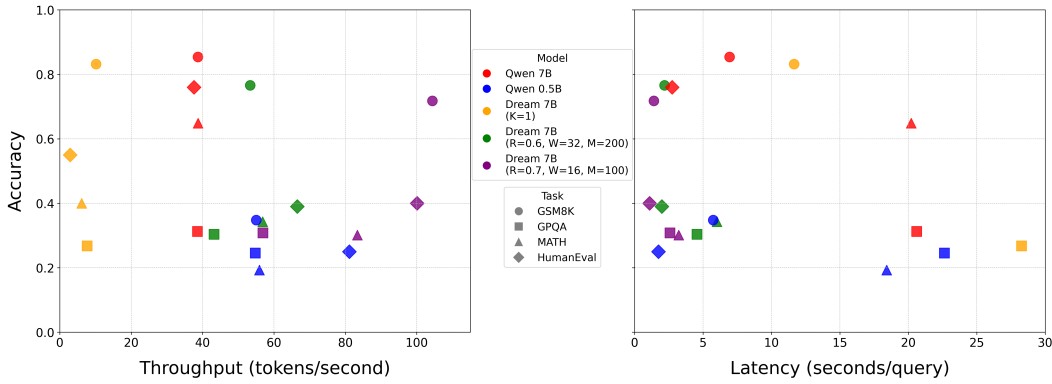

Figure 5: **The Pareto Frontier.** Dream 7B configured with ADP achieves substantially higher speeds with negligible impact on performance compared to the $K = 1$ base model. Notably, Dream with ADP surpasses the speed of autoregressive Qwen 7B and even Qwen 0.5B. This establishes Dream with ADP as Pareto-optimal, as no other model configuration is superior in both speed and quality. We show two hyperparameter configurations of APD, showing that we can flexibly tradeoff speed and quality.

another objective, such as increased latency. The Pareto frontier is, therefore, the collection of all such non-dominated configurations, representing the attainable boundary of performance.

In Figure 5, we show this frontier over several tasks and model configurations. Dream 7B ($K = 1$) is a naive baseline that decodes one token at a time, and it therefore occupies low throughput and high density regions. According to our previous assumptions, it will also upper bound the quality of Dream with other decoding parameters. We observe that Dream 7B with our configuration of parameters given by ADP, achieves much greater speed with minimal performance degradation from the base $K = 1$ performance. We also find that Dream with ADP is faster than the autoregressive Qwen 7B and even Qwen 0.5B. Note that if Qwen 7B used speculative decoding with a Qwen 0.5B draft model, it will still never exceed the throughput of ADP. Thus, Dream with ADP is Pareto-optimal, because no model can dominate it in speed and quality. Similar to throughput, ADP is also much faster in terms of latency. Diffusion models benefit from the fact that they generate fewer tokens to stay within a fixed context window, while autoregressive models can generate much longer reasoning traces that may ramble and not improve quality.

### 4.5 Qualitative Examples

Though we evaluate on standard benchmarks, APD can also quickly generate for more open-ended tasks. In Appendix D, we show examples with average number of parallelized tokens on a dataset of persuasive writing prompts [8].

## 5 Related Work

Several works propose architectural modifications to LLMs to enable multi-token prediction: Mask-Predict [12], Medusa [3], and DynaMo [37]. While these works are promising, our focus is multi-token prediction in dLLMs. Other recent works seek to improve dLLM speed. For example, block diffusion [2] enables KV caching, but requires training, unlike our method. Discrete copula diffusion [23] is an inference time method that aims to reduces the number of denoising steps required, but does not offer a tunable speed, quality tradeoff. More generally, algorithmic approaches to speeding up LLM inference include cascades [28], lookahead decoding [9], and speculative decoding [28], which are methods that are generally only applicable in autoregressive models. Building on speculative decoding, works have used different architectures as the draft model including any-order autoregressive models [14] and dLLMs [5]. We emphasize that using a large dLLM to draft in parallel presents fundamentally distinct challenges that are not solved with speculative decoding. Finally, we highlight that quantization is another promising and orthogonal approach for offering a strong speed quality tradeoff in LLMs [24, 7].

# 6 Conclusion

In this work, we pursue the significant challenge of increasing inference speed in large language models. We introduced Adaptive Parallel Decoding (APD), a novel algorithm that enables substantially faster sampling from dLLMs. APD uniquely restructures the dLLM into a left-to-right autoregressive process and leverages a smaller autoregressive model to assess the quality of parallel-generated token candidates. Complemented by optimizations such as KV caching and limited masked inputs, APD offers tunable parameters that allow a flexible trade-off between generation speed and output quality. This research offers a significant advancement in making dLLMs a more viable and efficient alternative for fast text generation.

# 7 Acknowledgment

Special thanks to Oliver Broadrick, who played an important role in brainstorming. This work was funded in part by the DARPA ANSR, CODORD, and SAFRON programs under awards FA8750-23-2-0004, HR00112590089, and HR00112530141, NSF grant IIS1943641, and gifts from Adobe Research, Cisco Research, and Amazon. We also received funding from NSF CAREER Grant #2341040, Schmidt Sciences Early Career Fellowship, and Okawa Foundation Research Award. Approved for public release; distribution is unlimited.

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

## A   Limitations

Adaptive Parallel Decoding offers a tradeoff, not a free lunch. Higher throughput will lead to lower quality, which is a limitation. We also do not claim to improve over the base diffusion model that we are sampling from. If Dream 7B is weak in a particular domain, our method will also perform poorly.

## B   Recompute KV Tradeoff

A major architectural difference between diffusion language models and autoregressive models is that diffusion models are trained with a bidirectional attention mask to predict masked tokens [17], and autoregressive models are trained to predict the next token with a causal mask. Because of this fundamental difference, autoregressive models have an invariant property: the attention matrix over a fixed set of tokens never changes over the course of a generation. Thus, previous "keys" and "values" (KV) used for computing attention can be cached, a process known as KV caching. While the same invariant property does not hold in diffusion language models, in practice, we can still perform KV caching. To adapt KV caching for diffusion language models, we relax the condition that every sampled token will immediately cache its keys and values. Instead we adopt a sliding window of size $W$ for which a token's KV will be recomputed. Once a sampled token is more than $W$ entries away from the most recently sampled tokens, we cache its KV.

Here, we show a modest gain in throughput for very minimal loss in accuracy by enabling KV caching in a diffusion model.

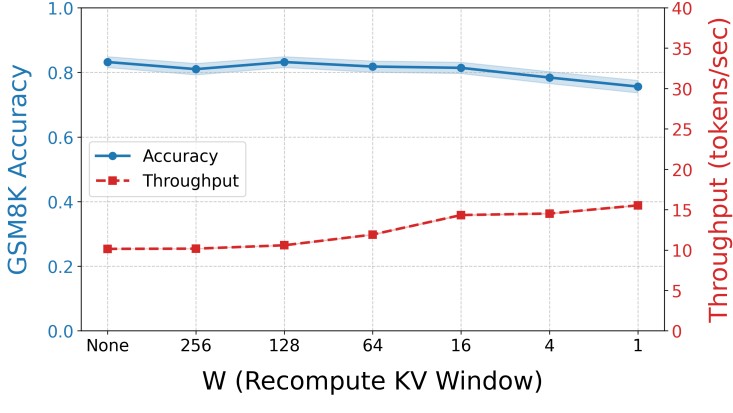

Figure 6

## C   Experimental Details

Because our models are instruction tuned, evaluate the model zero-shot with a chat format and system prompt. For GSM8K [6] and HumanEval [4], the standard system prompt is "You are a helpful assistant". For GPQA [33] and MATH [15], we modify the system prompt to specify the reasoning task and answer format.

Also, we set a maximum generation length of 256 or 512 tokens for the diffusion models and 16384 (the maximum context length) for the autoregressive Qwen models. This is a necessary difference, because diffusion models must materialize the maximum context length, unlike autoregressive models. In practice, the diffusion models generate [EOS] before the maximum generation length, at which point we end the generation.

When sampling from Dream 7B, we use the hyperparameters of temperature 0.2 and top-$p$ 0.95, as these are set as default.

# D   Qualitative Examples

Though we evaluate on standard benchmarks, APD can also quickly generate for more open-ended tasks. We show qualitative examples on a dataset of persuasive writing prompts [8] and sample from Dream 7B with APD parameter $R = 0.7$. With APD, we can perform well on open-ended tasks, but compared to reasoning, which has more textual regularities, we quantitatively do not observe the same rate of parallel generation as grade school math questions, for example.

---

**Prompt:**

Write a persuasive argument for the following claim: 'Gas car sales should be banned after a set phase-out date.'

---

**Answer:**

The use of gas cars is a major contributor to the emission of greenhouse gases, which are causing the climate change. Banning gas car sales after a set phase-out date would help to reduce the emission of greenhouse gases and the overall impact of climate change. Additionally, it would encourage the development of electric and hybrid vehicles, which are more environmentally friendly and help to reduce the reliance on fossil fuels.

---

*Average Parallel Tokens per Iteration: 3.38*

---

**Prompt:**

Write a persuasive argument for the following claim: 'Social media should verify user identities.'

---

**Answer:**

Social media platforms should verify user identities to ensure a safe and secure environment for all users. By verifying user identities, social media platforms can prevent the spread of fake accounts, cyberbullying, and other harmful activities. Additionally, verifying user identities can help to protect the privacy of users by ensuring that their personal information is not shared with unauthorized individuals.

---

*Average Parallel Tokens per Iteration: 2.92*

**Prompt:**

Write a persuasive argument for the following claim: 'Virtual emotional companion AI should be allowed and unrestricted'.

**Answer:**

Virtual emotional companion AI should be allowed and unrestricted because it can provide a valuable support system for individuals who are struggling with mental health issues. These AI can offer emotional support, companionship, and a sense of connection, which can be especially beneficial for those who are isolated or isolated. Additionally, virtual AI can provide a safe and non-judgmental space for individuals to express their feelings and thoughts, which can be especially helpful for those who are struggling with anxiety or depression. By allowing virtual emotional companion AI to be unrestricted, we can provide a support system for those who need it the most and help them to overcome their mental health challenges.

*Average Parallel Tokens per Iteration:* 2.88

