# OpenReview forum: "Accelerating Diffusion LLMs via Adaptive Parallel Decoding"
_NeurIPS.cc/2025/Conference — NeurIPS 2025 spotlight_

### Official Review · Reviewer_ezv3 · 2025-06-23

**Clarity:** 2
**Significance:** 3
**Originality:** 3
**Rating:** 4
**Confidence:** 4

**Summary:**

The authors propose Adaptive Parallel Decoding (APD), which improves sampling speed for existing diffusion-based large language models (dLLMs) compared to original methods like Dream and Llada. By introducing KV cache and limited masked inputs, APD enables flexible trade-offs between generation speed and quality.

**Questions:**

1. Llada’s reference paper tested more diverse datasets. Why restrict evaluation to only three?
2. The reference to "Dream 7B" (Citation 39) appears incorrect—no matching work was found. Is this a citation formatting error?

**Ethical Concerns:**

["NO or VERY MINOR ethics concerns only"]

**Final Justification:**

The author's rebuttal has basically resolved my question.

**Limitations:**

yes

**Quality:**

2

**Strengths And Weaknesses:**

Strengths:
1. APD adaptively determines the number of tokens generated per step via a small autoregressive model to evaluate output quality, mitigating the quality degradation caused by fixed-length parallel generation.
2. APD can be directly applied to existing dLLMs with significant speedup benefits.
3. The use of KV cache and limited masked inputs further enhances dLLM inference efficiency.

Weaknesses:
1. The proof relying on Bonferroni’s inequality (Property 1) is flawed. As noted in Line 122, $P_{APD}=P_{AR}$ holds only if $|g|=n$; when $|g|<n$, they are not equivalent. Thus, applying Bonferroni’s inequality is mathematically unsound. This also invalidates the heuristic justification for Property 2.
2. The author directly uses the KV cache method, but this method has already been proposed and there are different variants, so it cannot be considered an innovative point.
3.  Experiments are confined to GMS8K, GPQA, and MATH without theoretical guarantees, leaving generalization to other datasets unverified.

---

> ### Author Rebuttal · Authors · 2025-07-30
>
> Dear Reviewer ezv3,
>
> Thank you for your feedback. We hope to address your questions and concerns.
>
> >The proof relying on Bonferroni’s inequality (Property 1) is flawed. As noted in Line 122, $P_{APD}=P_{AR}$ holds only if $|g|=n$; when $|g|<n$, they are not equivalent. Thus, applying Bonferroni’s inequality is mathematically unsound. This also invalidates the heuristic justification for Property 2.
>
> We are not making a formal statement about $p\_{APD}$ or $\hat{p}\_{AR}$. Our proof shows that if $p\_D(x)=1$, then $p\_{AR}(x)=1$. Because the goal of $p_T$ is to approximate $p_{AR}$, we are stating that the same property must hold: if $p\_D(x)=1$, then $p\_{T}(x)=1$. This is our justification of Property 1.
>
> >The author directly uses the KV cache method, but this method has already been proposed and there are different variants, so it cannot be considered an innovative point.
>
> We are unsure which papers are being referred to here and respectfully ask for clarification. If the reviewer is referring to the use of KV cache more broadly for AR models, indeed there are many works which use it. KV caching in dLLMs is very natural in APD because we are decoding left to right, and the decoding order is similar to autoregressive. The challenge of KV caching in a dLLM is that because of bidirectional attention, previously computed keys and values can change. In practice, we show in the paper that only KV from a small window from the most recently generated tokens need to be updated to maintain performance. We do not consider KV caching the central point of the paper, but an optimization to achieve better results.
>
> >Experiments are confined to GMS8K, GPQA, and MATH without theoretical guarantees, leaving generalization to other datasets unverified.
>
> Because ADP exploits conditional dependence, which is a property of the data, to further parallel decoding, it is difficult to give theoretical guarantees on the speedups achievable. We emphasize that the advantage of ADP is that it dynamically adapts to the data to speed up decoding when it is possible without significantly degrading quality, but we are unable to formalize the precise impact on performance.
>
> >Llada’s reference paper tested more diverse datasets. Why restrict evaluation to only three?
>
> Many of the datasets for evaluating Llada such as MMLU and Hellaswag follow likelihood-based evaluation (eg, using multiple-choice questions) and are not generative, which is a requirement for evaluating our method. In light of the reviewers’ concern, we added a coding benchmark: Humaneval.
>
> | Model | HumanEval Accuracy | HumanEval Throughput |
> | :--- | :---: | :---: |
> | Qwen 7B | **0.76** | 37.6 |
> | Qwen 0.5B | 0.25 | 81.1 |
> | Dream 7B K=1 | 0.55 | 2.88 |
> | ADP Config 1 | 0.39 | 66.5 |
> | ADP Config 2 | 0.40 | **100.1** |
>
> **ADP Config 1**: Dream 7B + Qwen 2.5 0.5B, (R = 0.6, W = 32, M = 200)
>
> **ADP Config 2**: Dream 7B + Qwen 2.5 0.5B, (R = 0.7, W = 16, M = 100)
>
> As can be seen in this table, ADP works well for code. It maintains most of the accuracy of the underlying base model at K=1 but can achieve 34x the throughput of the base model. Additionally, it is firmly Pareto optimal by improving on Qwen 7B in throughput and Qwen 0.5B in quality.
>
> >The reference to "Dream 7B" (Citation 39) appears incorrect—no matching work was found. Is this a citation formatting error?
>
> We are citing the bibtex directly from the Dream 7B blog. There is currently no paper associated with the model.
>
> @misc{dream2025,
> title = {Dream 7B},
> url = {_omitting link due to rebuttal rules_},
> author = {Ye, Jiacheng and Xie, Zhihui and Zheng, Lin and Gao, Jiahui and Wu, Zirui and Jiang, Xin and Li, Zhenguo and Kong, Lingpeng},
> year = {2025}
> }
>
> We thank the reviewer for their valuable critiques, and we hope that our response provides enough assurance to increase our score.

---

> > ### Author Response · Authors · 2025-08-05
> > **Reminder**
> >
> > Hello Reviewer ezv3,
> >
> > We thank you again for your diligent review and feedback. We have tried our best to address your comments through the rebuttal period. Since the discussion is about to end, please let us know if there is anything that we can address.

---

> > > ### Comment · Reviewer_ezv3 · 2025-08-06
> > > **The response for authors' rebuttal**
> > >
> > > Thanks for the author's answer, which has basically resolved my question. I will increase my score to 4.

---

### Official Review · Reviewer_4Kag · 2025-06-27

**Clarity:** 3
**Significance:** 3
**Originality:** 3
**Rating:** 5
**Confidence:** 4

**Summary:**

This paper introduces Adaptive Parallel Decoding (APD), a novel method that accelerates diffusion-based LLMs by dynamically adjusting parallel token generation per decoding step. APD uses a multiplicative mixture between dLLM marginal probabilities and a small autoregressive model's joint distribution to determine token acceptance, enabling adaptive parallelization while maintaining quality. The framework integrates KV caching and masked lookahead constraints for acceleration and offers tunable parameters to balance throughput versus quality. APD achieves 59 tokens/sec (vs. 37 tokens/sec for autoregressive baselines) on reasoning tasks like GSM8K with minimal quality degradation, outperforming naive parallel approaches.

**Questions:**

Q1: How much latency does the small AR model introduce? Does KV caching sufficiently offset its computational overhead to justify inclusion?

Q2: Given Table 1 shows left-to-right decoding yields optimal dLLM performance, can APD effectively integrate with entropy/confidence-based decoding schemes (e.g., Llada’s approach) without compromising throughput gains?

Q3: Can APD achieve high parallelism (>5 tokens/step) in open-ended generation? Fig 2 suggests quality drops sharply for k>5 in naive parallelization, does APD overcome this limitation for non-reasoning domains?

**Ethical Concerns:**

["NO or VERY MINOR ethics concerns only"]

**Final Justification:**

In the paper, the authors propose APD to accelerate diffusion LLMs by dynamically adjusting parallel token generation per decoding step. The paper is interesting and the authors have addressed all the concerns from the reviewer. Hence, I give the paper a rating of 5.

**Limitations:**

Yes

**Quality:**

3

**Strengths And Weaknesses:**

Strengths:

+ This paper proposes the APD framework, by using a small AR model to validate large dLLM samples, dynamically adjust the number of parallel tokens, and solve the contradiction between parallelism and quality of dLLM.
+ The desigh integrates KV caching and mask lookahead restrictions, significantly accelerating dLLM inference without the need for re-training the model.
+ The effect is good. It achieves 59 tokens per second on tasks such as GSM8K, with a 60% increase in throughput and a quality loss of less than 2%.

Weaknesses:

- APD requires dLLMs specifically trained for left-to-right decoding, with non-left-to-right implementations such as Llada exhibiting substantially weaker performance as evidenced in Table 1.
- The small AR model adds latency; its impact on end-to-end throughput isn’t quantified.
- Evaluation focused on math/reasoning (GSM8K, GPQA, MATH). Open-ended generation shows lower parallelism (2.9–3.4 tokens/step) as evidenced in Appendix D, suggesting task-dependent efficacy.

---

> ### Author Rebuttal · Authors · 2025-07-30
>
> Dear Reviewer 4Kag,
>
> Thank you for your positive feedback. We hope to also answer your insightful questions and attend to your concerns.
>
> >APD requires dLLMs specifically trained for left-to-right decoding, with non-left-to-right implementations such as Llada exhibiting substantially weaker performance as evidenced in Table 1.
>
> To be clear, ADP does not *require* a dLLM trained for left to right decoding. Dream 7B’s training objective trains for random decoding, but because it was initialized from an autoregressive model it will have better representations in a left to right setting than Llada. However, Table 1 shows that Llada can also perform left to right decoding, and on GSM8K left to right decoding is even better than confidence based decoding. While ADP does not require any left to right training, the success of ADP does indicate that that parallel prediction can be performed left to right, and additional left-right fine tuning on top of dLLMs will likely improve the results of ADP.
>
> >Q1: How much latency does the small AR model introduce? Does KV caching sufficiently offset its computational overhead to justify inclusion?
>
> Thank you for the suggestion. We have compiled a table for GSM8K to show the impact of the small AR model.
>
> | Configuration | Latency (s) | AR Verification (s) | Ratio (%) |
> | :--- | :---: | :---: | :---: |
> | Dream K=1 | 11.6 | 0 | 0% |
> | ADP Config 1 | 2.18 | 1.09 | 50% |
> | ADP Config 2 | 1.40 | 0.73 | 52% |
>
> **ADP Config 1**: Dream 7B + Qwen 2.5 0.5B, (R = 0.6, W = 32, M = 200)
>
> **ADP Config 2**: Dream 7B + Qwen 2.5 0.5B, (R = 0.7, W = 16, M = 100)
>
> Overall, the latency using ADP is much lower than the default diffusion generation in Dream. However, the impact of the AR verification step is significant, and the model spends roughly half the time in this step. We see this as an opportunity for further optimizations. More efficient implementations of Gumbel sampling and better KV cache management using Paged Attention would likely reduce the latency significantly. The current implementation of ADP is purely PyTorch.
>
> >Q2: Given Table 1 shows left-to-right decoding yields optimal dLLM performance, can APD effectively integrate with entropy/confidence-based decoding schemes (e.g., Llada’s approach) without compromising throughput gains?
>
> As currently formulated, ADP can only work for left to right decoding. This is the case because the autoregressive verifier computes a left to right likelihood. Consider the following example where tokens are separated by commas
>
> >Prompt: The, capital, of
>
> >Diffusion Sample: France, is, is, is, is, is
>
> The diffusion model may sample this because it will be marginally likely due to there being many continuations besides "France" that can span multiple tokens. The autoregressive verifier will find the final “is” very likely due to the repetition, so we need to reject samples after the first disagreement between the diffusion samples and verifier (only accept up to the first “is”). This accept strategy produces a left to right decoding order. Adapting ADP for arbitrary decoding orders is an interesting open research question and may require a different auxiliary model such as an Any-Order Autoregressive model [1].
>
> >Q3: Can APD achieve high parallelism (>5 tokens/step) in open-ended generation? Fig 2 suggests quality drops sharply for k>5 in naive parallelization, does APD overcome this limitation for non-reasoning domains?
>
> The problem of parallel sampling is inherently probabilistic and therefore will be data dependent. It is likely that non-reasoning domains are less parallelizable in a fundamental way due to more interdependent tokens. Because open-ended generation is more difficult to evaluate, we do not include it in our benchmarks, but as Appendix D shows we can still achieve significant speed up over baseline diffusion models. The strength of ADP is that it can adaptively find opportunities to parallelize generation, and where this cannot be achieved, it automatically applies less parallelization so that quality is not severely impacted.
>
> [1] Shih, Andy, Dorsa Sadigh, and Stefano Ermon. "Training and inference on any-order autoregressive models the right way." Advances in Neural Information Processing Systems 35 (2022): 2762-2775.

---

### Official Review · Reviewer_mTc7 · 2025-07-01

**Clarity:** 3
**Significance:** 3
**Originality:** 4
**Rating:** 5
**Confidence:** 4

**Summary:**

LLMs today excel at text generation but remain bottlenecked by the strictly sequential nature of AR decoding. Diffusion-based LLMs offer parallel sampling in theory, yet in practice—exemplified by Dream and Llada—they only match AR quality when run one token at a time. To bridge this gap, this paper introduces APD: a left-to-right diffusion schedule that turns the model autoregressive, combined with a lightweight AR verifier to decide how many tokens to accept in parallel via a multiplicative product-of-experts criterion. They further accelerated inference with KV-caching beyond a sliding window and by setting a threshold for the masked lookahead. Together, these three tunable knobs let APD dramatically boost throughput—up to nearly twice the speed of pure AR—while incurring only minimal quality loss across three benchmarks.

**Questions:**

NA

**Ethical Concerns:**

["NO or VERY MINOR ethics concerns only"]

**Final Justification:**

This paper presents a significant advancement by introducing APD, a novel decoding strategy that achieves up to a 2x inference speed-up on challenging reasoning benchmarks while impressively maintaining near-autoregressive accuracy. The work provides a robust and practical solution to the inherent latency of diffusion-based LLMs, and while further generalization is future work, the existing results convincingly validate the method's core contribution to efficient text generation.

**Quality:**

3

**Strengths And Weaknesses:**

Strength:

1) Detailed analysis of the performance of the dLLMs in terms of accuracy and decoding strategies, showing the limitations of purely autoregressive generation and propose a novel method to maintain high quality while increasing the throughput.

2) Proposing enabling KV caching for tokens outside a pre-specified window together with maximum masked lookahead to increase the throughput of dLLMs.

3) APD consistently maintains near-autoregressive accuracy—even when accepting on average 5–7 tokens per iteration—while delivering up to a 2× speed-up over pure AR sampling on GSM8K, GPQA, and MATH.

Weaknesses:

1) I would appreciate a similar accuracy vs throughput comparison table similar to table 1, comparing the performance of Dream 7B + Qwen 2.5 0.5 with Qwen 7B to be able to observe the direct comparison of auto-regressive decoding and dLLMs with APD.

2) All experiments focus on three reasoning benchmarks (GSM8K, GPQA, MATH) and a single diffusion backbone (Dream 7B). Without additional evaluations—e.g., text summarization, code generation, translation—and tests on other dLLMs, it remains unclear whether APD’s speed-quality trade-offs generalize beyond the presented setup.

---

> ### Author Rebuttal · Authors · 2025-07-30
>
> Dear Reviewer mTc7,
>
> We are thankful that the reviewers recognize the novelty and impact of our work. We are also thankful for their insightful suggestions, which we aim to incorporate into the paper.
>
> As recommended, here is a table summary similar to Table 1 for the respective benchmarks:
>
>
> | Model | GSM8K Accuracy | GSM8K Throughput | GPQA Accuracy | GPQA Throughput | MATH Accuracy | MATH Throughput |
> | :--- | :---: | :---: | :---: | :---: | :---: | :---: |
> | Qwen 7B | **0.85** | 38.6 | **0.31** | 38.6 | **0.64** | 38.7 |
> | Qwen 0.5B | 0.34 | 55.0 | 0.24 | 54.7 | 0.19 | 55.8 |
> | Dream 7B K=1 | 0.83 | 10.1 | 0.26 | 7.66 | 0.40 | 6.15 |
> | ADP Config 1 | 0.76 | 53.3 | 0.30 | 43.2 | 0.34 | 56.7 |
> | ADP Config 2 | 0.71 | **104.3** | 0.30 | **56.9** | 0.30 | **83.3** |
>
> **ADP Config 1**: Dream 7B + Qwen 2.5 0.5B, (R = 0.6, W = 32, M = 200)
>
> **ADP Config 2**: Dream 7B + Qwen 2.5 0.5B, (R = 0.7, W = 16, M = 100)
>
> We will include these table results in the Appendix for more direct comparisons.
>
> To test for generalization to the other domains, we will also include coding in our evaluation suite. These results were obtained on NVIDIA A6000 GPUs due to memory errors, so they may not be directly comparable to the rest of the paper.
>
> | Model | HumanEval Accuracy | HumanEval Throughput |
> | :--- | :---: | :---: |
> | Qwen 7B | **0.76** | 37.6 |
> | Qwen 0.5B | 0.25 | 81.1 |
> | Dream 7B K=1 | 0.55 | 2.88 |
> | ADP Config 1 | 0.39 | 66.5 |
> | ADP Config 2 | 0.40 | **100.1** |
>
> As can be seen in this table, ADP results generalize beyond the existing benchmark scores. While it does not obtain the same quality as autoregressive due to a worse base model (as can be seen in the K=1 result), it is firmly on the Pareto frontier in terms of quality and throughput.
>
> We thank the reviewer for their generous feedback, and we hope that these updates have improved the paper significantly.

---

### Official Review · Reviewer_SxPY · 2025-07-05

**Clarity:** 3
**Significance:** 3
**Originality:** 3
**Rating:** 5
**Confidence:** 3

**Summary:**

This paper introduces Adaptive Parallel Decoding (APD), a novel decoding strategy designed to accelerate inference in diffusion-based large language models (dLLMs). Unlike traditional autoregressive decoding, which is inherently sequential, APD enables dynamic parallel token generation by combining the marginal probabilities from a diffusion model with the joint probabilities from a smaller auxiliary autoregressive model. The method introduces three tunable parameters—mixture weight (R), KV cache window (W), and maximum masked lookahead (M)—to balance the trade-off between speed and quality. Empirical results show that APD significantly improves throughput with minimal degradation in output quality, outperforming both naive diffusion decoding and autoregressive baselines on benchmarks like GSM8K, GPQA, and MATH.

**Questions:**

- In Table 1, why is random decoding always slower than entropy/confidence-base decoding for Dream/Llada models?

**Ethical Concerns:**

["NO or VERY MINOR ethics concerns only"]

**Limitations:**

The authors have addressed the limitations of their method.

**Quality:**

3

**Strengths And Weaknesses:**

Strengths:
- Paper is well motivated: Table 1 demonstrates that open source autoregressive LLMs still outperform the recent dLLMs in terms of performance v.s token/rate.
- Paper is well written and easy to read.
- The proposed technique is novel (as far as I know), the idea of adaptively choosing how many tokens to generate in parallel with a dLLM is new.
- The paper presents extensive empirical results on recent LLMs to back their claims.

Weaknesses:
- The dependence on an auxiliary approximate autoregressive model limits its applicability. At the very least, ADP requires that for each dLLM, a small LLM with the same tokenizer exists.
- I found sampling from $p_T$ as presented in eq (8) to be slightly confusing. In practice (as presented in Algorithm 1), $p_D$ is first used to generate $n$ tokens, which are then subsequently used by $\hat{p}_{AR}$. Does this not impose some conditional dependence?

---

> ### Author Rebuttal · Authors · 2025-07-30
>
> Dear Reviewer SxPY,
>
> We thank the reviewer for their positive feedback regarding our writing and empirical results. We hope to use this opportunity to clarify questions and concerns.
>
> >"The dependence on an auxiliary approximate autoregressive model limits its applicability. At the very least, ADP requires that for each dLLM, a small LLM with the same tokenizer exists."
>
> While ADP, as currently formulated, requires a smaller model with the same tokenizer, this problem is generally recognized to be the same in speculative decoding. However, a recently published work [1] shows that it is possible to perform speculative decoding over different tokenizers by performing verification over strings. This same idea naturally extends to ADP, where in our accept criterion, we can accept strings that the diffusion model and product mixture agree on.
>
> >"I found sampling from $p_T$ as presented in eq (8) to be slightly confusing. In practice (as presented in Algorithm 1), $p_D$ is first used to generate $n$ tokens, which are then subsequently used by $\hat{p}_{AR}$. Does this not impose some conditional dependence?"
>
> Intuitively, $p_D$ captures the marginals and $p_T$ can capture joint dependencies. We sample from $p_D$ initially because it can be performed in parallel. We then resample each token conditioned on the previous tokens in the sequence using $p_T$ and check that the original samples $p_D$ are valid under $p_T$. Thus, you are correct that $\hat{p}_{AR}$ (which is used to compute $p_T$) is adding conditional dependence between tokens. This is desirable because the inter-token dependencies are necessary to verify whether a sample should be accepted.
>
> >In Table 1, why is random decoding always slower than entropy/confidence-base decoding for Dream/Llada models?
>
> Thank you for the perceptive question. When a diffusion model outputs the EOS token with everything to its left sampled, the generation can stop. A random generation order is not likely to satisfy this condition early, while entropy/confidence/left-right are much more likely. Thus, generation order and answer length can change the throughput significantly, as the attention operation grows O(n^2) and the diffusion model is not KV caching by default.
>
> [1] Timor, Nadav, et al. "Accelerating llm inference with lossless speculative decoding algorithms for heterogeneous vocabularies." arXiv preprint arXiv:2502.05202 (2025).

---

### Decision · Program_Chairs · 2025-09-17

**Decision:**

Accept (spotlight)

**Comment:**

The paper presents a clear, useful decoding method for dLLMs with strong empirical throughput gains and modest quality costs, supported by additional analyses introduced during rebuttal. While broader task coverage, cross-tokenizer implementation, and stronger theory would further strengthen the work, the contribution is substantive and relevant for the community.